# To Build Our Future, We Must Know Our Past: Contextualizing Paradigm Shifts in Natural Language Processing

**Sireesh Gururaja**[1*]    **Amanda Bertsch**[1*]    **Clara Na**[1*]
**David Gray Widder**[2]    **Emma Strubell**[1,3]

[1]Language Technologies Institute, Carnegie Mellon University, Pittsburgh, PA, USA
[2]Digital Life Initiative, Cornell Tech, Cornell University, New York City, NY, USA
[3]Allen Institute for Artificial Intelligence, Seattle, WA, USA
{sgururaj, abertsch, csna, estrubel}@cs.cmu.edu, david.g.widder@gmail.com

## Abstract

NLP is in a period of disruptive change that is impacting our methodologies, funding sources, and public perception. In this work, we seek to understand how to shape our future by better understanding our past. We study factors that shape NLP as a field, including culture, incentives, and infrastructure by conducting long-form interviews with 26 NLP researchers of varying seniority, research area, institution, and social identity. Our interviewees identify cyclical patterns in the field, as well as new shifts without historical parallel, including changes in benchmark culture and software infrastructure. We complement this discussion with quantitative analysis of citation, authorship, and language use in the ACL Anthology over time. We conclude by discussing shared visions, concerns, and hopes for the future of NLP. We hope that this study of our field's past and present can prompt informed discussion of our community's implicit norms and more deliberate action to consciously shape the future.

## 1 Introduction

Natural language processing (NLP) is in a period of flux. The successes of deep neural networks and large language models (LLMs) in NLP coincides with a shift not only in the nature of our research questions and methodology, but also in the size and visibility of our field. Since the mid-2010s, the number of first-time authors publishing in the ACL Anthology has been increasing exponentially (Figure 1). Recent publicity around NLP technology, most notably ChatGPT, has brought our field into the public spotlight, with corresponding (over-)excitement and scrutiny.

In the 2022 NLP Community Metasurvey, many NLP practicioners expressed fears that private firms exert excessive influence on the field, that "a majority of the research being published in NLP is

*Denotes equal contribution.

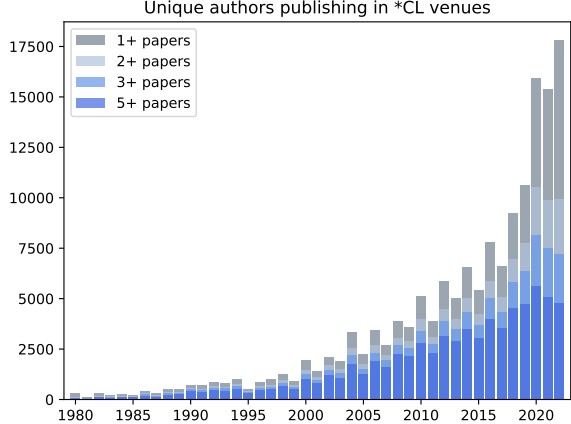

Figure 1: The number of unique researchers publishing in ACL venues has increased dramatically, from 715 unique authors in 1980 to 17,829 in 2022.

of dubious scientific value," and that AI technology could lead to a catastrophic event this century (Michael et al., 2022). More recently, there has been discussion of the increasing prevalence of closed-source models in NLP and how that will shape the field and its innovations (Rogers, 2023; Solaiman, 2023; Liao and Vaughan, 2023). In order to tackle these big challenges, we must understand the factors — norms, incentives, technology and culture — that led to our current crossroads.

We present a study of the community in its current state, informed by a series of long-form retrospective interviews with NLP researchers. Our interviewees identify patterns throughout the history of NLP, describing periods of research productivity and stagnation that recur over decades and appear at smaller scale around prominent methods (§3). Interviewees also point out unparalleled shifts in the community's norms and incentives. Aggregating trends across interviews, we identify key factors shaping these shifts, including the rise and persistence of benchmarking culture (§4) and the maturation and centralization of software infrastructure (§5). Our quantitative analysis of citation patterns, authorship, and language use in the ACL Anthology over time provides a complemen-

tary view of the shifts described by interviewees, grounding their narratives and our interpretation in measurable trends. Through our characterization of the current state of the NLP research community and the factors that have led us here, we aim to offer a foundation for informed reflection on the future that we as a community might wish to see.

## 2 Methods

### 2.1 Qualitative methods

We recruited 26 researchers to participate in interviews using *purposive* (Campbell et al., 2020) and *snowball* sampling (Parker et al., 2019): asking participants to recommend other candidates, and purposively selecting for diversity in affiliation, career stage, geographic position, and research area (see participant demographics in Appendix A.1). Our sample had a 69-31% academia-industry split, was 19% women, and 27% of participants identified as part of a minoritized group in the field. Of our academic participants, we had a near-even split of early-, mid-, and late-career researchers; industry researchers were 75% individual contributors and 25% managers.

Interviews were semi-structured (Weiss, 1995), including a dedicated notetaker, and recorded with participant consent, lasting between 45-73 minutes (mean: 58 minutes). Interviews followed an interview guide (see Appendix A.3) and began by contrasting the participant's experience of the NLP community at the start of their career and the current moment, then moved to discussion of shifts in the community during their career. Interviews were conducted between November 2022 and May 2023; these interviews were coincidentally contemporaneous with the release of ChatGPT in November 2022 and GPT-4 in March 2023, which frequently provided points of reflection for our participants.

Following procedures of grounded theory (Strauss and Corbin, 1990), the authors present for the interview produced analytical memos for early interviews (Glaser et al., 2004). As interviews proceeded, authors began a process of independently *open coding* the data, an *interpretive* (Lincoln et al., 2011) analytical process where researchers assign conceptual labels to segments of data (Strauss and Corbin, 1990). After this, authors convened to discuss their open codes, systematizing recurring themes and contrasts to construct a preliminary closed coding frame (Miles and Huberman, 1994). After this, an author who was not present in the interview applied the closed coding frame to the data. In weekly analysis meetings, new codes arose to capture new themes or provide greater specificity, in which case the closed coding scheme was revised, categories refined, and data re-coded in an iterative process. Analysis reported here emerged first from this coded data, was refined by subsequent review of raw transcripts to check context, and developed in discussion between all authors.

### 2.2 Quantitative Methods

We use quantitative methods primarily as a coherence check on our qualitative results. While our work is largely concerned with the *causes* and *community reception* of changes in the community, our quantitative analyses provide evidence that these changes have occurred. This includes analyzing authorship shifts (Figures 1 and 4), citation patterns (Figure 2), terminology use (Figure 2,3) in the ACL anthology over time; for more details on reproducing this analysis, see Appendix B.

## 3 Exploit-explore cycles of work

Our participants described cyclical behavior in NLP research following methodological shifts every few years. Many participants referred to these methodological shifts as "paradigm shifts", with similar structure, which we characterize as *explore* and *exploit* phases.[1]

**First wave: exploit.** Participants suggested that after a key paper, a wave of work is published that demonstrates the utility of that method across varying tasks or benchmarks. Interviewees variously describe this stage as *"following the bandwagon" (17)*, *"land grab stuff" (9)*, or *"picking the low-hanging fruit" (8)*. A researcher with prior experience in computer vision drew parallels between the rise of BERT and the computer vision community after AlexNet was released, where *"it felt like every other paper was, 'I have fine tuned ImageNet trained CNN on some new dataset' " (17)*. However, participants identified benefits of this *"initial wave of showing things work" (17)* in demonstrating the value of techniques across tasks or domains; in finding the seemingly obvious ideas which do *not* work, thus exposing new areas to investigate; and in developing downstream applications. When one researcher was asked to

---

[1] Inspired by the verbiage of reinforcement learning, e.g. Sutton and Barto (2018).

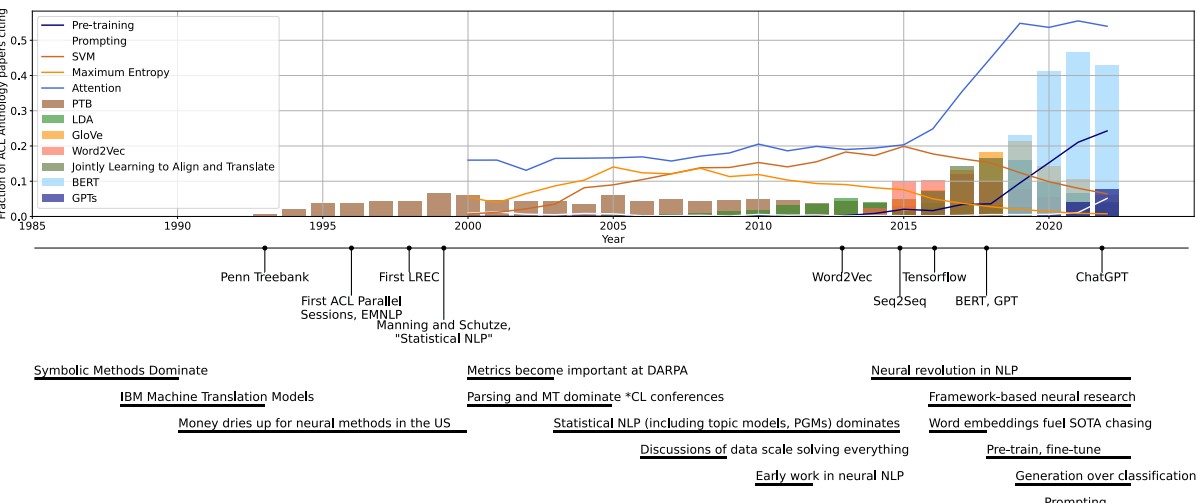

Figure 2: Quantitative and qualitative timeline. The lower half of this diagram captures historical information that our participants felt was relevant, along with their reported date ranges. The upper half captures quantitative information that parallels that timeline. Bar charts indicate fraction of papers that cite a given paper, while line charts indicate the fraction of papers that use a particular term.

identify their own exploit work, they joked we could simply *"sort [their] Google Scholar from most cited to least cited" (9)*, describing another incentive to publish the first paper applying a new methodology to a particular task. Participants additionally described doing exploit work early in their careers as a hedge against riskier, longer-term projects. However, most participants ascribed low status to exploit work[2], with participants calling these projects *"low difficulty" (4)* and *"obvious idea[s]" (9)* with *"high probability of success" (4)*. Participants also discussed increasing risk of being *"scooped" (7,9,4)* on exploit work as the community grows larger.

Participants felt that **the exploit phase of work is not sustainable.** Eventually, *"the low hanging fruit has been picked" (8)*; this style of work becomes unsurprising and less publishable. As one researcher put it: *"if I fine tune [BERT] for some new task, it's probably going to get some high accuracy. And that's fine. That's the end." (17)*.

**Second wave: explore**   After some time in the exploit phase, participants described a state where obvious extensions to the dominant methodology have already been proposed, and fewer papers demonstrate dramatic improvements over the state of the art on popular benchmarks.

In this phase, work on identifying the ways that the new method is flawed gains prominence. This

work may focus on interpretability, bias, data or compute efficiency. While some participants see this as a time of *"stalled" (8)* progress, others described this as *"the more interesting research after the initial wave of showing things work" (18)*. A mid-career participant identified this as the work they choose to focus on: *"I'm at the stage of my career where I don't want to just push numbers, you know. I'll let the grad students do that. I want to do interesting stuff" (22)*. Participants often saw "pushing numbers" as lower-status work, appropriate for graduate students and important for advancing the field and one's career, but ultimately not what researchers hope to explore.

Some work to improve benchmark performance was also perceived as explore work, particularly when it involved developing new architectures. One participant described a distinction between *"entering a race"* and *"forging in a new direction" (4)* with a project, which focuses the exploit/explore divide more on the perceived surprisingness of an idea rather than the type of contribution made. **Exploration often leads to a new breakthrough, causing the cycle to begin anew.**

### 3.1   Where are we now?

Placing the current state of the field along the exploit-explore cycle requires defining the current methodological "paradigm". Participants identified similar patterns at varying scales, with some disagreement on the timing of recent trends.

**Prompting as a methodological shift**   Several participants described prompting as a paradigm

---

[2]Data work is also commonly perceived as low-status (Sambasivan et al., 2021); participants agreed data work was previously undervalued in NLP but described a trend of increasing respect, one calling dataset curation *"valorized." (7)*

shift or a direction that the community found promising, but most participants viewed current work on prompt engineering or *"ChatGPT for X" (9)* as something that people are working on *"instead [...] of something that might make a fundamental difference" (14)*. One participant described both prompt engineering and previous work on feature engineering as *"psuedoscience [...] just poking at the model" (6)*. The current flurry of prompting work was viewed by several participants as lower-status work exploiting a known method.

**"Era of scale"** For participants who discussed larger-scale cycles, pre-trained models were frequently identified as the most recent methodological shift. Participants disagreed on whether scaling up pre-trained models (in terms of parameter count, training time, and/or pre-training data) was a form of exploiting or exploring this method. Some participants found current approaches to scale to be *"a reliable recipe where we, when we put more resources in, we get [...] more useful behavior and capabilities out" (4)* and relatively easy to perform: *"Once you have that GPU [...] it's like, super simple" (5)*. This perception of scaling up as both high likelihood of success and low difficulty places it as exploit work, and researchers who described scale in this way tended to view it as exploiting "obvious" trends. One researcher described scale as a way of establishing what is possible but *"actually a bad way to achieve our goals." (4)*, with further (explore-wave) work necessary to find efficient ways to achieve the same performance.

A minority of participants argued that, while historical efforts to scale models or extract large noisy corpora from the internet were exploit work, current efforts to scale are different, displaying *"emergence in addition to scale, whereas previously we just saw [...] diminishing returns" (22)*. Some participants also emphasized the engineering work required to scale models, saying that some were *"underestimating the amount of work that goes into training a large model" (8)* and identifying people or engineering teams as a major resource necessary to perform scaling work. The participants who described scaling work as producing surprising results and being higher difficulty also described scaling as higher status, more exploratory work.

**"Deep learning monoculture"** There was a sense from several participants that the current cycle has changed the field more than previous ones,

featuring greater centralization on fewer methods (see §5 for more discussion). Some expressed concern: *"a technique shows some promise, and then more people investigate it. That's perfectly appropriate and reasonable, but I think it happens a little too much. [...]* **Everybody collapses on this one approach [...] everything else gets abandoned."** *(19)*. Another participant described peers from linguistics departments who left NLP because they felt alienated by the focus on machine learning.

**Issues with peer review** Some felt that peer review was inherently biased toward incremental work because peer reviewers are invested in the success of the current methodological trends, with one participant arguing that *"if you want to break the paradigm and do something different, you're gonna get bad reviews, and that's fatal these days" (21)*. Far more commonly, participants did not express inherent opposition to peer review but raised concerns because of the recent expansion of the field, with one senior industry researcher remarking that peer reviewers are now primarily junior researchers who *"have not seen the effort that went into [earlier] papers" (12)*. Another participant asserted that *"**my peers never review my papers" (22)**. Participants additionally suggested that the pressure on junior researchers to publish more causes an acceleration in the pace of research and reinforcement of current norms, as research that is farther from current norms/methodologies requires higher upfront time investment.

This competitiveness can manifest in harsher reviews, and one participant described a *"deadly combination" (19)* of higher standards for papers and lower quality of reviews. Some participants described this as a reason they were choosing to engage less with NLP conferences; one industry researcher stated that *"I just find it difficult to publish papers in *CL [venues] that have ideas in them." (22)*.

## 4 Benchmarking culture

### 4.1 The rise of benchmarks

Senior and emeritus faculty shared a consistent recollection of the ACL community before the prominence of benchmarks as centralized around a few US institutions and characterized by *"patient money" (21)*: funding from DARPA that did not require any deliverables or statements of work. Capabilities in language technologies were show-

cased with technical *"toy" (26, 19)* demonstrations that were evaluated qualitatively: *"the performance metrics were, 'Oh my God it does that? No machine ever did that before.' " (21)*. Participants repeatedly mentioned how small the community was; at conferences, *"everybody knew each other. Everybody was conversing, in all the issues" (26)*. **The field was described as *"higher trust" (22)***, with social mediation of research quality – able to function in the absence of standardization because of the strong interconnectedness of the community.

Many participants recalled the rise of benchmarks in the late 1990s and early 2000s, coinciding with a major expansion in the NLP community in the wake of the "statistical revolution," where participants described statistical models displacing more traditional rules-based work (see Figure 2). In the words of one participant, the field became of *"such a big snowballing size that nobody owned the first evers anymore." (26)*. Instead, after the release of the Penn Treebank in 1993 and the reporting of initial results on the dataset, *"the climb started" (25)* to increase performance. Some participants attributed these changes to an influx of methods from machine learning and statistics, while others described them as methods to understand and organize progress when doing this coordination through one's social network was no longer feasible.

Over time, this focus on metrics seems to have overtaken the rest of the field, in part through the operationalization of metrics as a key condition of DARPA funding. One participant credited this to Anthony Tether, who became director of DARPA in 2001: they described earlier DARPA grants as funding *"the crazy [...] stuff that just might be a breakthrough" (21)* and DARPA under Tether as *"show me the metrics. We're going to run these metrics every year." (21)*.[3]

Some participants mourned the risk appetite of a culture that prioritized "first-evers," criticizing the lack of funding for ideas that did not immediately perform well at evaluations (notably leading to the recession of neural networks until 2011). However, there was sharp disagreement here; many other participants celebrated the introduction of benchmarks, with one stating that comparing results on benchmarks between methods *"really brought peo-*

---

[3]Other participants named DARPA program managers Charles Wayne and J. Allen Sears as additional key players. A recent tribute to Wayne (Church, 2018) provides additional context reflecting on DARPA's shift in priorities in the mid-1980s.

*ple together to exchange ideas. [...] I think this really helped the field to move forward." (2)*. Other participants similarly argued that a culture of quantitative measurement was key for moving on from techniques that were appealing for their *"elegance" (14)* but empirically underperforming.

## 4.2 The current state of benchmarks

Roughly twenty years on from the establishment of benchmarks as a field-wide priority, our participants' attitudes towards benchmarks had become significantly more complex. Many of our participants still found benchmarks necessary, but nearly all of them found them increasingly insufficient.

**Misaligned incentives** Many participants, particular early- and late-career faculty, argued that the field incentivizes the production of benchmark results to the exclusion of all else: *"the typical research paper...their immediate goal has to be to get another 2% and get the boldface black entry on the table." (21)*. For our participants, **improvements on benchmarks in NLP are the only results that are self-justifying to reviewers.** Some participants felt this encourages researchers to exploit modeling tricks to get state-of-the-art results on benchmarks, rather than explore the deeper mechanisms by which models function (see §3).

**"We're solving NLP"** Some participants perceive a degradation in the value of benchmarks because of the strength of newer models. Participants appreciated both the increased diversity and frequency of new benchmark introduction, but noted that the time for new approaches to reach *"superhuman" (6,22)* levels of performance on any specific benchmark is shortening. One common comparison was between part of speech tagging (*"a hill that was climbed for [...] about 20 years" (15)*) and most modern benchmarks ("solved" within a few years, or even months). Some went further, describing *"solving NLP" (8)* or naming 2020 as the time when *"classification was solved" (15)*.

However, when participants were asked for clarification on what it meant to "solve" a problem, most participants hedged in similar ways; that datasets and benchmarks could be solved, with the correct scoping, but problems could rarely or never be solved. Many participants argued that the standard for solving a task should be human equivalency, and that this was not possible without new benchmarks, metrics, or task definition.

**NLP in the wild** Some participants argued that many benchmarks reflect tasks that *"aren't that useful in the world" (13)*, and that this has led to a situation where *"[NLP], a field that, like fundamentally, is about something about people, knows remarkably little about people" (3)*. Industry participants often viewed this as a distinction between their work and the academic community, with one stating that *"most of the academic benchmarks out there are not real tasks" (12)*. Many academics articulated a desire for more human-centered NLP, and most participants described feeling pressure over the unprecedented level of outside interest in the field. One participant contrasted the international attention on ChatGPT with the visibility of earlier NLP work: *"It's not like anyone ever went to like parser.yahoo.com to run a parser on something" (3)*. Participants argued that, given this outside attention, the benchmark focus of NLP is too narrow, and that **benchmarks fail to capture notions of language understanding that translate to wider audiences**, and that we should move on from benchmarks not when they are saturated but when *"it wouldn't really improve the world to improve this performance anymore" (9)*. This echoed a common refrain: many participants, especially early- and mid-career researchers, saw positive social change as a goal of progress in NLP.

## 5 Software lotteries

Hooker (2021) argues that machine learning research has been shaped by a *hardware lottery*: an idea's success is partially tied to its suitability for available hardware. Several participants spoke about software in ways that indicate an analogous *software lottery* in NLP research: as the community centralizes in its software use, it also centralizes in methodology, with researchers' choices of methods influenced by relative ease of implementation. This appeared to be a relatively new phenomenon; participants described previously using custom-designed software from their own research group, or writing code from scratch for each new project.

**Centralization on frameworks** As deep learning became more popular in NLP, participants described the landscape shifting. As TensorFlow (Abadi et al., 2015) increased support for NLP modeling, PyTorch (Paszke et al., 2019) was released, along with NLP-specific toolkits such as DyNet (Neubig et al., 2017) and AllenNLP (Gardner et al., 2018), and *"everything started being [...]*

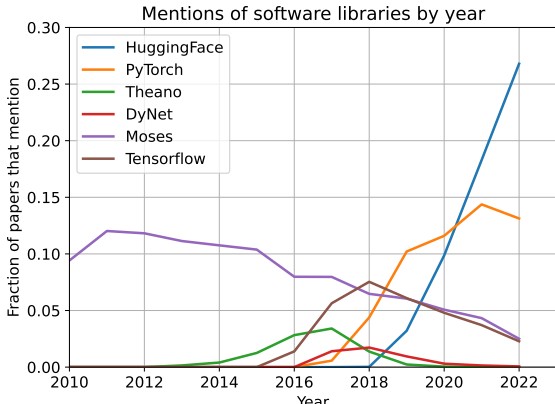

Figure 3: Mentions of libraries over time in the ACL Anthology. Note the cyclic pattern and increasing concentration on the dominant framework over time. While some libraries are built on others, the shift in mentions over time captures the primary level of abstraction that researchers consider important. See appendix B for details on how we handle ambiguity in mentions.

*simpler to manage, simpler to train" (17)*. Previously, participants described spending *"like 90% of our time re-implementing papers" (12)*; as more papers began releasing code implemented in popular frameworks, the cost of using those methods as baselines decreased. One participant stated that *"things that software makes easy, people are going to do" (18)*; this further compounds **centralization onto the most popular libraries, with little incentive to stray from the mainstream**: *"everybody uses PyTorch, so now I use PyTorch too" (8)*; *"we just use HuggingFace for pretty much everything" (18).*[4] Figure 3 visualizes mentions of frameworks across papers in the ACL Anthology, showing both the rise and fall in their popularity. The rising peaks of popularity reflect the centralization over time. While some communities within NLP had previously seen some centralization on toolkits (e.g. machine translation's use of Moses (Koehn et al., 2007)), the current centralization transcends subfields.

**Centralization on specific models** Participants identified another shift after the release of BERT and subsequent development of Hugging Face. Because of pre-training, participants moved from merely using the same libraries to *"everyone us[ing] the same base models" (9)*. Participants expressed concern that this led to further centraliza-

---

[4]In this section, we primarily discuss open source frameworks commonly used by academic and industry researchers. However, many of our participants working in industry also describe affordances and constraints of *internal* toolkits that are used at their respective companies.

tion in the community, with one participant identifying a trend of some people *"not us[ing] anything else than BERT [...] that's a problem" (5)*. This concentration around particular modeling choices has reached **greater heights than any previous concentration on a method**; in 2021, 46.7% of papers in the ACL anthology cited BERT (Devlin et al. (2019); Figure 2)[5].

Other large models are only available to researchers via an API. One participant who works on LLMs in industry argued that black-box LLMs, while *"non-scientific" (15)* in many ways, were like large-scale technical tools constructed in other disciplines, drawing a parallel to particle physics: *"computer science is getting to have its CERN moment now [...] there's only one Large Hadron Collider, right?" (15)*. This participant argued that NLP has become a field whose frontiers require tools that are beyond most organizations' resources and capabilities to construct, but nonetheless are widely adopted and set bounding parameters for future research as they see wide adoption. In this vision, black-box LLMs take on the same role as the LHC or the Hubble Space Telescope (both notably public endeavors, unlike most LLMs), as tools whose specifications are decided on by a few well-resourced organizations who shape significant parts of the field. But most participants expressed skepticism about the scientific validity of experiments on black-box LLMs, with one participant referencing critiques of early-2000s IR research on Google (Kilgarriff, 2007).

**Centralization on Python** While most early-career and late-career participants did not express strong opinions about programming languages, many mid-career participants expressed strong dislike for Python, describing it as *"a horrible language" (22)* with efficiency issues that are *"an impediment to us getting things done" (20)* and data structure implementations that are *"a complete disaster in terms of memory" (9)*. One participant described their ideal next paradigm shift in the field as a shift away from using Python for NLP.
Yet even the participants who most vehemently opposed Python used it for most of their research,

citing the lack of well-supported NLP toolkits or community use in other languages. This is an instance of a software lottery at a higher level, where the dominance of a single programming language has snowballed with the continued development of research artifacts in that language.

## 5.1 Consequences of centralization

This increasing centralization of the modern NLP stack has several consequences. One of the primary ones, however, is the loss of control of design decisions for the majority of researchers in the community. Practically, researchers can now choose from a handful of well-established implementations, but only have access to software and models once the decisions on how to build them have already been reified in ways that are difficult to change.

**Lower barriers** Beyond time saved (re-) implementing methods, many participants identified a lower barrier to entry into the field as a notable benefit of centralization on specific software infrastructure. Participants described students getting state of the art results within an hour of tackling a problem; seeing the average startup time for new students decreasing from six months to a few weeks; and teaching students with no computer science background to build NLP applications with prompting.

**Obscuring what's "under the hood"** One participant recalled trying to convince their earlier students to implement things from scratch in order to understand all the details of the method, but no longer doing so because *"I don't think it's possible [...] it's just too complicated" (11)*; others attributed this to speed more than complexity, stating that *"the pace is so fast that there is no time to properly document, there is no time to properly engage with this code, you're just using them directly" (5)*. However, this can cause issues on an operational level; several participants recalled instances where a bug or poor documentation of shared software tools resulted in invalid research results. One participant described using a widely shared piece of evaluation code that made an unstated assumption about the input data format, leading to *"massively inflated evaluation numbers" (3)* on a well-cited dataset. Another participant described working on a paper where they realized, an hour before the paper deadline, that the student authors had used two different tokenizers in the pipeline by mistake: *"we decided that well, the results were still valid, and the results would only get better if [it was fixed]...so*

---

[5]While not every paper that cites BERT uses a BERT model, this indicates how central BERT is both as a model and as a frame for the discussion of other work. For comparison, only two other papers have been cited by more than 20% of anthology papers in a single year: "Attention is All You Need" (Vaswani et al., 2017) with 27% in 2021 and GloVe (Pennington et al., 2014) with 21% in 2019.

*the paper went out. It was published that way." (26)* Software bugs in research code are not a new problem,[6] but participants described bugs in toolkits as difficult to diagnose because they *"trust that the library is correct most of the time" (8)*, even as they spoke of finding **"many, many, many" (8) bugs in toolkits** including HuggingFace and PyTorch.

**Software is implicit funding**  Participants suggested that tools that win the software lottery act as a sort of implicit funding: they enable research groups to conduct work that would not be possible in the tools' absence, and many of our participants asserted that the scope of their projects expanded as a result. However, they also significantly raise the relative cost of doing research that does not fall neatly into existing tools' purview. As one participant stated, *"You're not gonna just build your own system that's gonna compete on these major benchmarks yourself. You have to start [with] the infrastructure that is there" (19)*. This is true even of putatively "open" large language models, which do not necessarily decentralize power, and can often entrench it instead (Widder et al., 2023). This set of incentives pushes researchers to follow current methodological practice, and some participants feared this led toward more incremental work.

### 5.2  Impact on Reproducibility

A common sentiment among participants was that centralization has had an overall positive impact on reproducibility, because using shared tools makes it easier to evaluate and use others' research code. However, participants also expressed concerns that the increasing secrecy of industry research complicates that overall narrative: *"things are more open, reproducible... except for those tech companies who share nothing" (14)*.

**Shifts in expectations**  One participant described a general shift in focus to *"making sure that you make claims that are supported rather than reproducing prior work exactly" (12)* in order to match reviewers' shifting expectations. However, participants also felt that the expectations for baselines had increased: *"[in the past,] everybody knew that the Google system was better because they were running on the entire Internet. But like that was not a requirement [to] match Google's accuracy. But now it is, right?" (8)*.

---

[6]Tambon et al. (2023) describe *silent bugs* in popular deep learning frameworks that escape notice due to undetected error propagation.

**Disparities in compute access**  Many felt that building large-scale systems was increasingly out of reach for academics, echoing concerns previously described by Ahmed and Wahed (2020). Participants worried that *"we are building an upper class of AI" (6)* where most researchers must *"build on top of [large models]" (15)* that they cannot replicate, though others expressed optimism that *"clever people who are motivated to solve these problems" (22)* will develop new efficient methods (Bartoldson et al., 2023). Industry participants from large tech companies also felt resource-constrained: *"modern machine learning expands to fit the available compute." (4)*.

## 6  Related Work

The shifts we explore in this paper have not happened in a vacuum, with adjacent research communities such as computer vision (CV) and machine learning (ML) experiencing similar phenomena, inspiring a number of recent papers discussing norms in AI more broadly. Birhane et al. (2022) analyze a set of the most highly cited papers at recent ML conferences, finding that they eschew discussion of societal need and negative potential, instead emphasizing a limited set of values benefiting relatively few entities. Others have noticed that corporate interests have played an increasing role in shaping research, and quantified this with studies of author affiliations over time in machine learning (Ahmed and Wahed, 2020) and NLP (Abdalla et al., 2023). Su and Crandall (2021) study the tangible emotional impact of recent dramatic growth in the CV community by asking community members to write stories about emotional events they experienced as members of their research community.

While we focus on summarizing and synthesizing the views of our participants, some of the overarching themes identified in this work have been discussed more critically. Fishman and Hancox-Li (2022) critique the unification of ML research around transformer models on both epistemic and ethical grounds. Position papers have critiqued the notion of general purpose benchmarks for AI (Raji et al., 2021), and emphasized the importance of more careful and deliberate data curation in NLP (Rogers, 2021; Bowman and Dahl, 2021).

The NLP Community Metasurvey (Michael et al., 2022) provides a complementary view to this work, with their survey eliciting opinions from a broad swath of the *CL community on a set of 32

controversial statements related to the field. The survey also asked respondents to guess at what the most popular beliefs would be, eliciting sociological beliefs about the community. While there is no direct overlap between our questions and Metasurvey questions, participants raised the topics of scaling up, benchmarking culture, anonymous peer review, and the role of industry research, which were the subject of Metasurvey questions. Where we can map between our thematic analysis and Metasurvey questions, we see agreement– e.g. many of our participants discussed others valuing scale, but few placed high value themselves on scaling up as a research contribution.

The availability of the ACL Anthology has enabled quantitative studies of our community via patterns of citation, authorship, and language use over time. Anderson et al. (2012) perform a topic model analysis over the Anthology to identify different eras of research and better understand how they develop over time, and analyze factors leading authors to join and leave the community. Mohammad (2020) analyze citation patterns in *CL conferences across research topics and paper types, and Singh et al. (2023) specifically inspect the phenomenon wherein more recent papers are less likely to cite older work. Pramanick et al. (2023) provide a view of paradigm shifts in the NLP community complementary to ours based on a diachronic analysis of the ACL Anthology, inferring causal links between datasets, methods, tasks and metrics.

Shifts in norms and methods in science more broadly has been studied outside computing-related fields. Most notably, Kuhn (1970) coined the term *paradigm shift* in *The Structure of Scientific Revolutions*. His theory of the cyclic process of science over decades or centuries has some parallels with the (shorter timescale) exploit-explore cycles discussed in this work. Note that in this work, we did not prime participants with an *a priori* definition of paradigm shift, allowing each participant to engage with the term according to their own interpretation, which often differed from Kuhn's notion of a paradigm shift.

## 7 The Future

The rise of large language models has coincided with disruptive change in NLP: accelerating centralization of software and methodologies, questioning of the value of benchmarks, unprecedented public scrutiny of the field, and dramatic growth of the community. A shift like this can feel threatening to the fundamental nature of NLP research, but this is not the first period of flux in the field, nor are the fundamental forces enabling LLMs' dominance and other changes entirely new.

Our participants described cycles of change in the NLP community from mid-80s to the present, with common themes of first exploiting and then exploring promising methodologies. Each methodological shift brought corresponding cultural change: the shift from symbolic to statistical methods brought about the rise of benchmark culture and the end of the socially mediated, small-network ACL community. Neural methods began the centralization on software toolkits and the methodologies they support. Pre-training intensified this software lottery, causing unprecedented levels of centralization on individual methods and models. Current models have called into question the value of benchmarks and catapulted NLP into the public eye. Our participants largely agree on the resulting incentives– to beat benchmark results, to do the easiest thing rather than the most fulfilling, to produce work faster and faster – while largely expressing frustration with the consequences.

We hope that this contextualization of the current state of NLP will both serve to inform newer members of the community and stir informed discussion on the condition of the field. While we do not prescribe specific solutions, some topics of discussion emerge from the themes of this work:

- Who holds the power to shape the field? How can a broad range of voices be heard?
- Do the incentives in place encourage the behavior we would like to see? How can we improve reviewing to align with our values?
- What affects the ability to do longer-term work that may deviate from current norms?
- How can the community arrive at an actively mediated consensus, rather than passively being shaped by forces like the ones we discuss?

We personally take great hope for our community from this project. The care with which all participants reflected on the shape of the field suggests to us that many people are concerned about these issues, invested in the community, and hopeful for the future. By sharing publicly what people so thoughtfully articulate privately, we hope to prompt further discussion of what the community can do to build our future.

## Limitations

**Western bias**   The most notably *irrepresentative* sampling bias in our participant pool is the lack of non-Western institutional affiliation (and the strong skew toward North American affiliations). This bias has arisen likely in part due to our own institutional affiliation and conceptions of the community. That being said, given the Association for Computational Linguistics' historically US- and English-centric skews, this allows us to gather historical perspectives. Additionally, considering that Western institutions constitute a citation network largely distinct from Asian networks (Rungta et al., 2022), we believe that our sample allows us to tell a rich and thorough story of factors which have shaped the Western NLP research community, which both informs and is informed by other communities of NLP researchers.

**Lack of early career voices**   Our inclusion criteria for our participants– three or more publications in *CL, IR, or speech venues[7]– necessarily means that we have limited perspectives on and from more junior NLP researchers (such as junior graduate students), those hoping to conduct NLP research in the future, those who have engaged with NLP research in the past and decided not to continue before developing a publication record, and those who have consciously decided *not* to engage with NLP research in the first place. In general, although we gathered perspectives from participants across a variety of demographic backgrounds, our participants represent those who have been successful and persisted in the field. This is especially true for our participants in academia; of our participants' current academic affiliations, only R1 institutions (if in the US) and institutions of comparable research output (if outside the US) are represented. We therefore may be missing perspectives from certain groups of researchers, including those who primarily engage with undergraduate students or face more limited resource constraints than most of the academic faculty we interviewed.

Future research could further examine differences between geographic subcommunities in NLP and more closely examine influences on people's participation in and disengagement from the community. Additionally, we leave to future work a more intentional exploration of perspectives from early career researchers and those who have not yet published but are interested in NLP research.

## Ethics Statement

Following Institutional Review Board recommendations, we take steps to preserve the anonymity of our participants, including aggregating or generalizing across demographic information, avoiding the typical practice of providing a table of per-interviewee demographics, using discretion to redact or not report quotes that may be identifying, and randomizing participant numbers. Participants consented to the interview and to being quoted anonymously in this work. This work underwent additional IRB screening for interviewing participants in GDPR-protected zones.

We view our work as having potential for positive impact on the ACL community, as we prompt its members to engage in active reflection. We believe that, given recent developments in the field and the co-occuring external scrutiny, the current moment is a particularly appropriate time for such reflection. Additionally, we hope that our work can serve those currently *external* to the community as an accessible, human-centered survey of the field and factors that have shaped it over the decades, prioritizing sharing of anecdotes and other in-group knowledge that may be difficult for outsiders to learn about otherwise.

## Acknowledgements

This work would not have been possible without our participants, who were generous with their time and engaged deeply with this material. We would additionally like to thank Jill Fain Lehman, Anjalie Field, Dawn Nafus, Momin M. Malik, Graham Neubig and the anonymous reviewers for their helpful comments.

This work was supported in part by a grant from the National Science Foundation Graduate Research Fellowship Program under Grant No. DGE2140739. Widder gratefully acknowledges the support of the Digital Life Initiative at Cornell Tech. Any opinions, findings, and conclusions or recommendations expressed in this material are those of the authors and do not necessarily reflect the views of the sponsors.

---

[7]In order to capture perspectives of the community changing over time, and to select for people who are part of these communities.

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

## A Details of Qualitative Methodologies

### A.1 Participant demographics

Of the academics interviewed, there was an even split between early, mid, and late career (6/33% each). Of those in industry, 6 (75%) were individual contributors and 2 (25%) were research managers. Our sample is only 19% women, which is likely representative, as women comprise approximately 15% of tenure-track computer science faculty in the US (Way et al., 2016). For more discussion of the sample characteristics, see 7. Our positive response rate was 82%.

| Demographic trait | % of sample |
|---|---|
| In academia | 69% |
| Women | 19% |
| Minoritized group | 27% |
| Born outside US | 38% |
| Currently works outside US | 4% |

Table 1: Self-reported demographic makeup of subjects.

### A.2 Consent protocol

Interviewees were asked for verbal consent unless they were in a GDPR-protected region at the time of the interview, in which case they provided written consent via DocuSign instead. This consent script was IRB-approved.

> Hi, thanks for taking the time to talk with me! My collaborators on this project and I work for CMU. We can be reached at [emails] should you have any questions for us during the study or after.
>
> This interview will take between 45 minutes and an hour. There will be no compensation for participation.
>
> Participation is always voluntary, and you may refuse to participate in the research study or stop participation at any time.
>
> You will not be identified in any reports we release from this research. This data will be deidentified, which means you will not be identified by name or any other specific characteristic. We may quote you anonymously. Just in case, though, please do not reveal any private or personally-identifiable information about yourself or others in your answer to our questions.
>
> I'd like to record the audio of this interview as a memory aid. You can ask me to stop the recording at any time. Only the members of our research team will have access to these recordings.
>
> We really appreciate your participation, and we hope to publish this research to advance our understanding of the factors shaping the NLP research community.
>
> Do you have any questions? If not, do I have your consent to participate in this study?
>
> [answer any questions; if consented, begin recording]
>
> Alright, I've started the recording. Just to confirm, do I have your consent to participate in this study, and to record this interview?

### A.3 Interview Guide

These questions are intentionally open-ended, and the interviewers asked non-scripted followups or additional questions where appropriate. Over time, as early themes emerged, additional questions were added, particularly on funding and pace of work.

1. First, I have a few questions about your relationship to the NLP community. You can be as specific or as vague as you'd like with your responses.

    (a) What do you consider to be your main/home/primary research community?

    (b) More specifically, what venues do you follow and/or publish in?

    (c) What subarea(s) or subfields are you most active in?

    (d) Is this different from what you have considered your main community at other points in your career? (Prompt: if so, what changed?)

    (e) What would you define as the start of your NLP research career? (e.g. start of PhD, research as an undergrad, etc). (Prompt: When was this?)

2. I'd like to hear your thoughts on what the field was like near the beginning of your career.

    (a) When you started in your field, what did people generally think were the most promising directions? (Prompt: do you agree?)

    (b) How do you interpret the term "promising"?

    (c) What do you think the research community prioritized when you started?

    (d) What was the scope of the work that your research group did?

    (e) What was your relationship to computing resources at the start of your career?

    (f) What did your software workflow look like when you first started doing research? (Prompt: What tools, frameworks, libraries did you use?)

    (g) Where did funding for your work come from? (Prompt: what were the major costs involved with your research?)

    Now, I'd like to compare this with the current state of the field.

    (a) What do you think others in your field would say are the most promising directions? (Prompt: do you agree?)

    (b) What do you think the research community prioritizes now?

    (c) What is the scope of the work that your research group does?

    (d) How do computing resources affect your group's work now?

    (e) How does the software workflow look like for you or your students now?

    (f) What tools, frameworks, libraries do you or your students use?

    (g) Have these tools, frameworks, and libraries made an impact on your (or your students') research?

    (h) What impacts do you think these tools, frameworks, and libraries have made on your community's research?

    (i) Where does the funding for your work come from? (Prompt: what are the major costs involved in your research?)

    (j) When you or your students start a new project, how long on average do you expect it to take, from the start of work to a paper submission?

    (k) Has this changed over your career? (Prompt: what has led to this change?)

    Now I'd like to hear your thoughts on the changes you've observed in your career.

    (a) Are there paradigm shifts that you would identify in the field over the course of your career?

    (b) Did your community change at all, as a result of these paradigm shifts?

    (c) Prompt: what years would you ascribe to each shift?

(d) How frequently do you feel the community undergoes a paradigm shift? (Prompt: is this frequency changing?)

(e) Are there concerns you have with the direction of the field?

(f) If there were to be a paradigm shift in the next few years, in what direction should it go?

(g) Do you think changes in the research community have changed your teaching? (Prompts: how? How do you feel about this shift?)

Now I have some demographic questions, which will help us understand the range of people that we talk to. If you'd prefer not to answer any of them, just let me know.

(a) With which gender do you identify?
   i. Man
   ii. Woman
   iii. Or, feel free to specify as you wish

(b) I am going to read some age brackets. Can you indicate when I read a bracket that your age falls into?
   i. 18-24
   ii. 25-34
   iii. 35-44
   iv. 45-54
   v. 55-64
   vi. 65+

(c) Which country were you born in?

(d) (if not already known) Which country are you currently based in?

(e) What stage of your academic career would you consider yourself in?

(f) Do you consider yourself a part of a minoritized group in your field?

(g) Anything else in your background that feels relevant or that you want to add?

3. Finally, we'd like to hear from more people about these issues. Is there anyone you could introduce us to who you think would have interesting answers to these questions?

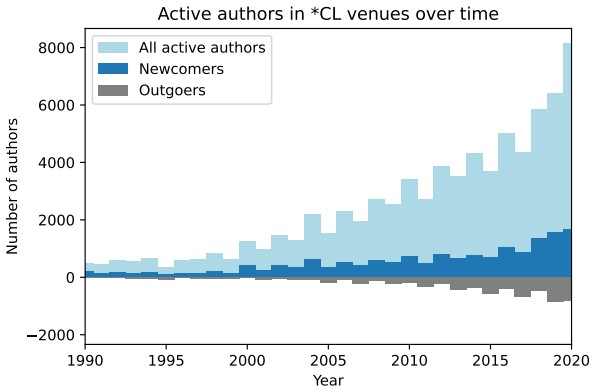

Figure 4: The number of "active" researchers publishing in ACL venues has increased dramatically, with more newcomers to the field year over year

## B    Detailed quantitative methodology

We begin with the ACL Anthology and focus on papers between 1980 and 2022. Using the Semantic-Scholar (S2) API (Wade, 2022), we obtain author and citation information of papers indexed by S2 (some venues, such as some workshops, and some Findings[8] papers, are not indexed, leaving $77,235$ papers), with a focus on citations of papers identified by our participants as having been influential to the NLP community. We select from this set of influential papers to generate the bar plots for Figure 2. Figure 1 uses author publication information linked to the individual papers considered.

We rely on S2ORC (Lo et al., 2020) for full-text PDF parses of a subset of these papers, which we use match for mentions of software toolkits identified by our participants as having been influential to the community, for Figure 3, as well as for mentions of influential techniques in the line plot of Figure 2. Note that quantifying mentions is noisy: framework/library names can be spelled in a variety of ways, and names like "Moses" are also used for authors in the ACL Anthology. For figure 3, we normalize by lowercasing all text, and using the most common normalized spelling, e.g. `tensorflow`, or `huggingface`. We estimate that this will overestimate the presence of Moses, due to its other usages. and underestimate the presence of Hugging Face, which is officially spelled "Hugging Face", but much more often used as "HuggingFace". Despite this, the incredible growth and popularity of Hugging Face relative to other frameworks is still prominently visible.

We present an alternative view of data, similar to that seen in Figure 1, in Figure 4. Here, we define members of the community as authors with at least three papers total in *CL venues. Authors are counted as "leaving" the community the year after their last *CL publication. We only consider trends in authorship until 2020, as it is difficult to determine if authors who did not publish in the last few years have left the community indefinitely.

---

[8]As of October 2023, while some Findings papers, such as from ACL 2021 and EMNLP 2020, are automatically indexed by S2, Findings papers from some conferences such as EMNLP 2022 are not. While some of these papers (or versions of them) may still be indexed by S2 due to also being on ArXiv or a similar service, we do not include them in our set of papers.