# OpenReview forum: "To Build Our Future, We Must Know Our Past: Contextualizing Paradigm Shifts in Natural Language Processing"
_EMNLP/2023/Conference — EMNLP 2023 Main_

### Official Review · Reviewer_tGxC · 2023-08-04

**Soundness:** 4

**Excitement:**

4: Strong: This paper deepens the understanding of some phenomenon or lowers the barriers to an existing research direction.

**Paper Topic And Main Contributions:**

This work is a sociological study of the field of NLP as currently perceived by its practitioners, primarily based on the long-form interviews of a sample of 26 researchers and supplemented by qualitative longitudinal analysis of citation, terminology and authorship patterns in ACL anthology. A formal protocol was established to sample the participants, conduct and code the interviews.

The author(s) then provide a qualitative analysis of the various topics of the interviews, describing NLP researcher's self reflections on the history of the field (introducing the idea of recurrent cycles of exploration and exploitation), and also of the particularly strong pull that the latest breakthroughs produced, potentially leading to a "deep learning monoculture". A perception of the fundamental shift from symbolic to statistical methods is described, along with deep methodological and cultural changes, such as the rise of "benchmarking culture" and the dispersion of a community that used to be small and highly connected. Another topic is the centralization not only of methods but also of tools, where not long ago most code was written from scratch or maintained at the lab level, now a few libraries such as PyTorch came to dominate. This trend is qualitatively confirmed with ACL Anthology data. Researcher reflect on the "software lottery" effect, where methods are more likely to be studied if they happen to match the capabilities of the dominant software, another phenomenon likely to hurt diversity. These now ubiquitous libraries are also described as "implicit funding", leveraging what can be achieved by a lab. The current level of competition in the field, dominance in numbers of junior researchers (also at the peer review level) and heavy exploitation phase around transformers are perceived by some researchers to make it very hard for original ideas to actually be published, and create a career risk in pursuing them.

**Reasons To Accept:**

I greatly enjoyed reading this article. I am not a sociologist, but having collaborated with them I have some awareness of the methodology. The sampling strategy, the way the interviews were conducted and the coding protocol appear sound and rigorous to me. I also appreciate the effort to validate conclusions as much as possible with quantitative methods on the corpus of publications.

The gold nugget in my view is the qualitative analysis of the interviews. It is clear, lucid and well-written. It touches on all the fundamental topics that I perceive to be crucial to reflect about at this point: the history and cycles of the field, its research and publication practices, the need to reward more novelty and risk, the direct and also complex possible consequences of a model and software mono-culture, benchmarking culture (its pros and cons), the growing influence of private companies, the close models, the rise of "prompt engineering", the "CERN" era of NLP. I can't think of one topic that preoccupied me in the last year in relation to NLP that is not covered.

In short: this work is great and the timing is more than appropriate. I believe that its publication will be a good service to the community.

**Reasons To Reject:**

I see no reason to reject this article.

**Reproducibility:**

N/A: Doesn't apply, since the paper does not include empirical results.

**Reviewer Confidence:**

4: Quite sure. I tried to check the important points carefully. It's unlikely, though conceivable, that I missed something that should affect my ratings.

---

> ### Author Rebuttal · Authors · 2023-08-28
>
> Thank you for taking the time to review our paper and for your kind words! We are glad to hear that you appreciate the rigor and timeliness of our work and share in our excitement about its potential impact. We will be happy to address any questions in the discussion.

---

### Official Review · Reviewer_nTbt · 2023-08-05

**Typos Grammar Style And Presentation Improvements:** Figure 3 isn't referenced in the main…
**Soundness:** 1

**Ethical Concerns:**

Yes

**Excitement:**

2: Mediocre: This paper makes marginal contributions (vs non-contemporaneous work), so I would rather not see it in the conference.

**Justification For Ethical Concerns:**

I feel generalizing voices of 26 NLP researchers over a field of thousands can create a bias in the minds of junior researchers or new researchers trying to enter the field.

Disclaimer: This is a personal opinion.

**Paper Topic And Main Contributions:**

The paper aims to document a qualitative survey of 26 NLP researchers over the history of NLP in terms of Explore-Exploit nature of studies, the current state, the benchmarking culture, and the software availability.
The authors try to depict certain historical timelines with the quantitative measures of papers, citations etc. in ACL anthology.
The author's state their motivation as driving the future of NLP by knowing the past.

**Reasons To Accept:**

I like:
1) (Briefly) discussing some of the issues of the current state of NLP. For ex., issues with scale and the rat race of publishing.


**Reasons To Reject:**

The paper would better fit as a technical/interviews report.
1) Most of the paper seems anecdotal relying on the interviews from 26 anonymized NLP researchers.
2) The paper doesn't focus on LLMs but a general thought process of how the 26 NLP researchers have viewed NLP since the time they entered into the field v/s the current state. For ex., issues with peer reviews.
3) The paper isolates only prompting as a methodology whereas there are many other strategies employed currently. For ex., zero shot/ few shot, fine-tuning only the classification layer, adapters etc.


Side: The paper compares the trends of softwares but includes toolkits and libraries in the same graph (which is not the right comparison). For ex., HuggingFace is a toolkit that is built using Pytorch/Tensorflow, and the authors compare Pytorch, Tensorflow usage to HuggingFace.
Also, figure 3 that makes this comparison is not referenced in the main text.

**Reproducibility:**

N/A: Doesn't apply, since the paper does not include empirical results.

**Reviewer Confidence:**

4: Quite sure. I tried to check the important points carefully. It's unlikely, though conceivable, that I missed something that should affect my ratings.

---

> ### Author Rebuttal · Authors · 2023-08-28
>
> Thank you for taking the time to review our paper! Please see our response below.
>
> > **Most of the paper seems anecdotal relying on the interviews from 26 anonymized NLP researchers.**
>
> Our primary methodology is qualitative. While qualitative methods originated and have robust grounding in fields as diverse as sociology, psychology and anthropology, they have long been used in NLP and related fields such as HCI and linguistics. Some recent examples of qualitative research conducted primarily via long-form interviews include work from NAACL 2022 [1], FAccT 2023 [2], and one of the best papers from CHI 2023 [3].
>
> We followed norms from qualitative methods and grounded theory to ensure that we carried out a rigorous study, including a deliberate and justified sample, carefully constructed and administered interview guide, and established data coding and analysis methods. 26 participants is well within the norm for this kind of work. We took care to collect a representative sample across multiple dimensions of identity, and we discuss the ways in which we interpret our results in light of our sample characteristics in our Limitations section.
>
> Like all methods, interview methods have limitations, some of which are remedied by our complementary quantitative methods (section 2.2), and the remaining of which we carefully document in the limitations section. We welcome pointers to any more specific limitations we may have omitted. We are confident that our methods yield insights that would be difficult to uncover otherwise, including understandings of researchers’ thought processes that cannot be explored through quantitative methods or short-form surveys.
>
> [1] Kaitlyn Zhou, Su Lin Blodgett, Adam Trischler, Hal Daumé III, Kaheer Suleman, and Alexandra Olteanu. 2022. Deconstructing NLG Evaluation: Evaluation Practices, Assumptions, and Their Implications. In Proceedings of the 2022 Conference of the North American Chapter of the Association for Computational Linguistics: Human Language Technologies, pages 314–324, Seattle, United States. Association for Computational Linguistics.
>
> [2] Wesley Hanwen Deng, Nur Yildirim, Monica Chang, Motahhare Eslami, Kenneth Holstein, and Michael Madaio. 2023. Investigating Practices and Opportunities for Cross-functional Collaboration around AI Fairness in Industry Practice. In Proceedings of the 2023 ACM Conference on Fairness, Accountability, and Transparency (FAccT '23). Association for Computing Machinery, New York, NY, USA, 705–716. https://doi.org/10.1145/3593013.3594037
>
> [3] Lauren Wilcox, Renee Shelby, Rajesh Veeraraghavan, Oliver L. Haimson, Gabriela Cruz Erickson, Michael Turken, and Rebecca Gulotta. 2023. Infrastructuring Care: How Trans and Non-Binary People Meet Health and Well-Being Needs through Technology. In Proceedings of the 2023 CHI Conference on Human Factors in Computing Systems (CHI '23). Association for Computing Machinery, New York, NY, USA, Article 489, 1–17. https://doi.org/10.1145/3544548.3581040
>
>
> > **The paper doesn't focus on LLMs but a general thought process of how the 26 NLP researchers have viewed NLP since the time they entered into the field v/s the current state. For ex., issues with peer reviews.**
>
> While current discussion in the field is centered around LLMs and the disruptive changes they have brought, our work contextualizes our present and future by analyzing the historical precedent of this shift. LLMs are only one of many other historical “paradigm shifts”, which we uncover, analyze and learn from in this work. Our analysis shows that cultural shifts, such as growing discontent with peer review, emerge alongside methodological shifts– e.g. with statistical methods and benchmarking, and with neural methods and increased software centralization. Thus, we believe discussing current perceptions of the field both culturally and methodologically is appropriate for the theme track. For instance, participants attribute issues with peer review partially to the burgeoning size of the community; the community’s growth is a consequence of the growing public perception of the field, which in turn was catalyzed by LLMs leading to more public-facing NLP applications. One of the contributions of our work is uncovering and analyzing these connections, which we will update the next version of the paper to clarify and highlight.
>
> Please see our response to Reviewer bhU3’s Question 1 for additional context.
>
> > **The paper isolates only prompting as a methodology whereas there are many other strategies employed currently. For ex., zero shot/ few shot, fine-tuning only the classification layer, adapters etc.**
>
> To avoid biasing interview respondents, we do not *a priori* identify methodologies of note– we discuss methodologies that interviewees themselves identify as major shifts or otherwise bring up organically when discussing changes in the field. While one or two interviewees mentioned these other methods, nearly all of our interviewees discussed prompting, suggesting that prompting is dominating the conversation around recent methodologies.
>
> > **Comparing HuggingFace and Pytorch/Tensorflow usage isn’t the right comparison because HuggingFace is built using Pytorch/Tensorflow**
>
> While it’s true that using HuggingFace requires also using Pytorch/Tensorflow, Figure 3 compares explicit references to each toolkit/library in the text of papers. The intuition here is that people generally reference the toolkit/library they are using most directly, rather than all software they use (in the same way that people who use PyTorch directly do not generally also write that they’re using Python). This is not to say that people who discuss using HuggingFace are not using a PyTorch/Tensorflow backend.
>
> This comparison is useful because the abstractions that HuggingFace provides are different from using PyTorch directly. We discuss the benefits and drawbacks of increasing software abstraction in Section 5. We will clarify this in the caption of Figure 3 and in discussion of this figure.
>
> > **Figure 3 isn't referenced in the main text.**
>
> Thanks! We will correct this in the next version.
>
>
> > **I feel generalizing voices of 26 NLP researchers over a field of thousands can create a bias in the minds of junior researchers or new researchers trying to enter the field.**
>
> This is a thoughtful concern. We hope that the measures we took can alleviate some of these potential risks. We took care to describe the limitations and sampling biases of our study explicitly (see our Limitations section), as well as highlight both agreements and disagreements between our participants. We welcome feedback calling attention to any specific additional limitations, blind spots, or potential negative consequences of our study.
>
> It is also (perhaps increasingly) true that new and junior researchers are entering the field with their own preconceptions about it. We are sympathetic to the many stresses and pressures the community’s junior researchers may be experiencing, due to factors such as recent developments in research, media coverage of AI/NLP in general, and competitive admissions to graduate programs. We believe that our work has greater potential to help new and junior researchers *expand* their own perspectives by increasing awareness of perspectives from other “eras,” subfields, and institutions, than to bias them in any specific way. From our ethics statement:
>
> > We view our work as having potential for positive impact on the ACL community, as we prompt its members to engage in active reflection. We believe that, given recent developments in the field and the co-occuring external scrutiny, the current moment is a particularly appropriate time for such reflection. Additionally, we hope that our work can serve those currently *external* to the community as an accessible, human-centered survey of the field and factors that have shaped it over the decades, prioritizing sharing of anecdotes and other in-group knowledge that may be difficult for outsiders to learn about otherwise.

---

### Official Review · Reviewer_bhU3 · 2023-08-12

**Typos Grammar Style And Presentation Improvements:** Line 181
**Soundness:** 4

**Excitement:**

4: Strong: This paper deepens the understanding of some phenomenon or lowers the barriers to an existing research direction.

**Missing References:**

N/A

**Paper Topic And Main Contributions:**

This work is a community study that provides a concrete review of the history in the landscape of NLP, from the perspectives of norms, incentives, technology, and culture that have contributed to its present state. The methodology involves in-depth interviews with 26 NLP researchers of varied backgrounds, and their insights are thoughtfully categorized into various pertinent themes. Additionally, the study includes a quantitative assessment of citations, authorship, and terminology usage within ACL conferences, designed to enhance the ongoing discourse. It appears that many of the discussions within the paper are predominantly centered around Western institutions and funding structures, potentially introducing a significant bias. To the authors' credit, they do acknowledge this issue in their limitations section, and the value of their work remains intact.


**Questions For The Authors:**

Question 1: Reflecting on the theme, what roles do large language models play in shaping the current state of NLP?

Question 2: Among the challenges highlighted in this paper, which ones are unique to the field of NLP, and which ones extend to the broader AI community?

Question 3: Can you provide some specific actions that various stakeholders could undertake based on the findings of this study?


**Reasons To Accept:**

Strength 1: This community study is solid and beautifully written. The qualitative methodologies adhere to ethical standards and appropriate protocols for interactions with human subjects. The presentation of information is clear and thought-provoking.

Strength 2: This paper is remarkably timely, arriving at a juncture where NLP is undergoing substantial transformation. The crescendo of voices and sentiments surrounding the success and hype of large language models is undeniable. While this study refrains from offering definitive predictions about the future, it adeptly highlights numerous deeply entrenched issues. The paper's impact is readily foreseen in its role of raising public awareness about several concerns within the NLP community.


**Reasons To Reject:**

While I resonate with the sentiments expressed in this paper, which reflect some of the concerns I share, I'd like to draw the authors' attention to a few limitations that I've identified.

Weakness 1: Neglect of practitioners' viewpoints: I was surprised to find that the topics covered in this paper largely revolve around research management matters, such as funding and resources. I noticed a lack of mention regarding the growing challenge of reproducing research papers and establishing fair comparisons between baselines. This raises concerns that the study participants might primarily consist of research managers in industry or faculty members in academia, rather than NLP practitioners or students deeply involved in research implementation.

Weakness 2: Absence of novice voices: To enhance its comprehensiveness, this work could benefit from including more perspectives from beginners and first-time researchers. Many students within this community face even more challenging conditions, with restricted access to computational resources and limited opportunities to secure research assistant roles within established research groups (which are crucial if they need recommendation letters for higher education applications).

Weakness 3: Absence of tangible action items. While this work effectively raises public awareness regarding numerous concerns within the NLP community, it appears to lack a concrete list of actionable steps. What are some venues to trigger these discussions? How can current conference and reviewing systems improve to at least take initiative? The absence of such discourse left me somewhat disappointed.


**Reproducibility:**

4: Could mostly reproduce the results, but there may be some variation because of sample variance or minor variations in their interpretation of the protocol or method.

**Reviewer Confidence:**

4: Quite sure. I tried to check the important points carefully. It's unlikely, though conceivable, that I missed something that should affect my ratings.

---

> ### Author Rebuttal · Authors · 2023-08-28
>
> Thank you for taking the time to review our paper and for your kind words! We are glad to hear that you find our work well-written, thought-provoking, and timely, and that you anticipate it to be impactful. Please see our response below.
>
> > **Neglect of practitioners' viewpoints**
>
> > **lack of mention regarding the growing challenge of reproducing research papers and establishing fair comparisons between baselines**
>
> Our least senior academic interviewees were post-docs and first-year faculty, and only one of our industry interviewees was in a management position, as opposed to a hands-on, IC role. That is to say, a significant number of our interviewees are practitioners.
> Participants generally leaned toward discussing factors like funding and resource availability; this may be because we framed our questions to focus on the factors that shape research more broadly, which may have led our respondents to take a more high-level view.
>
> We touch on reproducibility in Section 5.2, although there we focus on reproducibility concerns arising from disparities in compute access. Due to the page limit, we cut a previous section on reproducibility more broadly. A common sentiment was that converging on similar tools and frameworks for research has benefited reproducibility overall, although participants expressed concerns that large models and increasing secrecy by tech companies were harming reproducibility. Some participants also tied this to a shift in peer review, feeling that  reviewers’ expectations overall have shifted away from valuing reproducibility.
> To address the reviewer’s concern, we will add back discussion of these concerns in the next version of the paper.
>
>
> > **Absence of novice voices**
>
> This is a great point, and one that we discussed in the final stages of this project. Our framing was focused on the factors shaping the field, both now and historically, and as such, we wanted to ask people that tend to have higher leverage than graduate students - professors who advise labs and industry researchers with experience. We would like to do follow up work here about the experience of more junior researchers – and there is some excellent prior work (e.g. [1]) – but we did not feel that interviewing junior researchers naturally fit into the scope of this paper.
>
> [1] Ignat, Oana, Zhijing Jin, Artem Abzaliev, Laura Biester, Santiago Castro, Naihao Deng, Xinyi Gao, Aylin Gunal, Jacky He, Ashkan Kazemi, Muhammad Khalifa, Nam Ho Koh, Andrew Lee, Siyang Liu, Do June Min, Shinka Mori, Joan Nwatu, Verónica Pérez-Rosas, Siqi Shen, Zekun Wang, Winston Wu and Rada Mihalcea. “A PhD Student's Perspective on Research in NLP in the Era of Very Large Language Models.” ArXiv abs/2305.12544 (2023).
>
>
> > **Absence of tangible action items**
>
> > **Question 3: Can you provide some specific actions that various stakeholders could undertake based on the findings of this study?**
>
> This is another great point – and one that we debated ourselves. We share the reviewer’s sentiment in being hopeful for positive change, though we intentionally decided not to prescribe any particular solutions to our potential audience, especially given the absence of junior researchers in our sample. Instead, we came up with specific *questions* surfaced by our research that we believe the NLP community would benefit from considering. These questions direct action and reflection to points of controversy without prescribing any specific path forward. Though we shortened these significantly for length in Section 7, we have decided to include an expanded list in the next version.
>
> Additional questions we will raise in the expanded discussion:
> > 1. What is the role of benchmarks when the community has exploded in size and “NLP kinda works now”? What alternatives might we embrace in addition to existing prevalent methods of evaluation? How can we embrace more holistic evaluations of models, despite the difficulty of scaling them? How can conference organizers set guidelines for reviewing to help align evaluation of work with the things we value?
> > 2. Do the incentives that our field has in place encourage the behavior we'd like to see? What kinds of people and research are valued by those incentives? What kind of work has higher and lower status?
> > 3. How do we feel about the pace and pressure of NLP research today? How can we encourage people to do longer-term work that may deviate from current norms?
> > 4. Even among our sample of 26, there were substantial disagreements. How do we move forward as a community? How do we wish to define the community and the relationships between its subcommunities?
>
>
> > **Question 1: Reflecting on the theme, what roles do large language models play in shaping the current state of NLP?**
>
> This was one of the questions that we started with in this work, but in the process of conducting the interviews, we realized that while language models appear to be the driving force of change in NLP today, several of the cultural changes that we attribute solely to language models arise from the way NLP developed through the 2010s more broadly. The increasing pace of work and burgeoning community size, for instance, especially seen in relation to language model research, is fed by the centralization of methods, software and design decisions that we highlight.
>
> However, LLMs have intensified these trends, and several of the trends that we see now. In particular, the increasing public perception of the field now that NLP “kinda works” and the centralization around a few particular models because of their size and the difficulty in reproducing them are linked directly to language models.
>
> Each major methodological shift (e.g. statistical NLP, neural NLP, pretrained models) that our interviewees discussed also heralded cultural shifts. The ways that we react as a community to current concerns– e.g. peer review, centralization, and compute disparities–  will define the future cultural impact of LLMs. We believe that this makes this the perfect time to reflect on previous shifts in the community, and to learn from how we responded to those shifts.
>
> We will update the next version of the paper to clarify and highlight these connections.
>
>
> > **Question 2: Among the challenges highlighted in this paper, which ones are unique to the field of NLP, and which ones extend to the broader AI community?**
>
> This is a great question– while we don’t have evidence to make strong claims here, some interviewees did discuss their experiences in NLP in comparison to CV, speech, and ML.
> Some commonalities identified by participants include the public visibility of the umbrella category of “AI” (e.g. the common gaze on things like self-driving cars and LLMs), the cycles of work after breakthroughs, and concerns over peer review– although participants disagreed on the scale of NLP’s challenges relative to those of related communities. Another point of reflection was *where* NLP placed in a broader community– some participants referred to NLP as a “subfield of ML/AI,” while others drew strong distinctions between ML as a method for NLP and as a central point of the community. We will add some discussion of this in the next version.

---

### Meta-Review · Area_Chair_THs8 · 2023-09-19

**Recommendation:** 4

**Metareview:**

This paper presents interviews with 26 NLP practitioners about the current state of the field. While the sample size is small, the study was conducted following proper protocols and included in-depth qualitative analyses of the various topics in the interview, supplemented with qualitative longitudinal analyses of citations, terminology, and authorship patterns in the ACL Anthology.

---

### Decision · Program_Chairs · 2023-10-07

**Decision:**

Accept-Main

**Comment:**

This paper presents interviews with 26 NLP practitioners about the current state of the field. While the sample size is small, the study was conducted following proper protocols and included in-depth qualitative analyses of the various topics in the interview, supplemented with qualitative longitudinal analyses of citations, terminology, and authorship patterns in the ACL Anthology.